# Image Super-Resolution Reconstruction Network Based on Structural Reparameterization and Feature Reuse

**DOI:** 10.3390/s25195989

**Published:** 2025-09-27

**Authors:** Tianyu Li, Xiaoshi Jin, Qiang Liu, Xi Liu

**Affiliations:** 1School of Information Science and Engineering, Shenyang University of Technology, Shenyang 110870, China; litianyu@sut.edu.cn (T.L.); liuxi@sut.edu.cn (X.L.); 2Engineering Training Center, Shenyang University of Technology, Shenyang 110870, China; liuqiangcc@sut.edu.cn

**Keywords:** image super-resolution reconstruction, structural reparameterization, feature reuse

## Abstract

In the task of integrated circuit micrograph acquisition, image super-resolution reconstruction technology can significantly enhance acquisition efficiency. With the advancement of deep learning techniques, the performance of image super-resolution reconstruction networks has improved markedly, but their demand for inference device memory has also increased substantially, greatly limiting their practical application in engineering and deployment on resource-constrained devices. Against this backdrop, we designed image super-resolution reconstruction networks based on feature reuse and structural reparameterization techniques, ensuring that the networks maintain reconstruction performance while being more suitable for deployment in resource-limited environments. Traditional image super-resolution reconstruction networks often redundantly compute similar features through standard convolution operations, leading to significant computational resource wastage. By employing low-cost operations, we replaced some redundant features with those generated from the inherent characteristics of the image and designed a reparameterization layer using structural reparameterization techniques. Building upon local feature fusion and local residual learning, we developed two efficient deep feature extraction modules, and forming the image super-resolution reconstruction networks. Compared to performance-oriented image super-resolution reconstruction networks (e.g., DRCT), our network reduces algorithm parameters by 84.5% and shortens inference time by 49.8%. In comparison with lightweight image reconstruction algorithms, our method improves the mean structural similarity index by 3.24%. Experimental results demonstrate that the image super-resolution reconstruction network based on feature reuse and structural reparameterization achieves an excellent balance between network performance and complexity.

## 1. Introduction

Image analysis and quantitative interpretation capabilities in material characterization [1], biological monitoring [2], and defect inspection [3] have been significantly advanced by the deep fusion of computer vision with microscopic imaging, thereby expediting developments in these fields. Microscopic imaging is established as a key inspection modality in integrated-circuit (IC) manufacturing, packaging, and post-silicon verification. In the manufacturing phase, it is employed for evaluating pattern quality after lithographic development [4], detecting film residues and defects subsequent to etching [5]. In the packaging and post-silicon verification phases, it is utilized to inspect die warpage, cracks, and contamination after bonding, and it supplies high-resolution images for hardware-Trojan localization and confirmation [6]. With increasing chip area and shrinking transistor feature sizes, the data volume and physical resolution demanded for large-area microscopic imaging at a given spatial resolution have escalated sharply. Limited by diffraction, electron-beam aberrations, and the inherent trade-off between field of view and resolution, the throughput of existing microscopic systems constitutes the principal bottleneck to acquiring high-resolution chip images, posing severe challenges to imaging timeliness [7].

Image super-resolution reconstruction involves generating a high-resolution (HR) image with enhanced details from one or multiple low-resolution (LR) images. During large-area imaging tasks such as wafer defect scanning, overall throughput is markedly improved by super-resolution through a workflow that reduces physical scan counts, decreases data volume, and expedites subsequent processing [8]. Escalating resolution requirements in IC inspection cause the physical dimensions and cost of high-resolution CMOS/CCD sensors to increase sharply with pixel density, along with elevated noise; these hardware limitations are compensated by super-resolution algorithms [7]. Resolution breakthroughs and cost optimization in IC microscopic imaging are jointly realized via algorithmic innovation and hardware co-design.

As a classic yet challenging low-level vision task, image super-resolution reconstruction is inherently an ill-posed problem. Thanks to the progress in efficient computing hardware and advanced algorithms [9], deep learning has demonstrated remarkable capabilities in processing large amounts of unstructured data. Consequently, deep learning-based approaches for image super-resolution reconstruction have gained significant momentum. Lim et al. proposed Enhanced Deep Super-Resolution (EDSR) [10], which adopted a residual backbone and removed the batch normalization (BN) layers, in addition to regularly increasing the depth of the network, significantly increased the number of output features in each layer. Zhang et al. introduced the residual dense network (RDN) [11], which utilized residual dense blocks to extract rich local features through densely connected convolutional layers in the thesis. SwinIR [12] is an image super-resolution network based on the Swin Transformer. It leverages the hierarchical structure and sliding window mechanism of the Swin Transformer to effectively process images and generate high-resolution outputs. HAT (Hybrid Attention Transformer) [13] combines channel attention and self-attention mechanisms. This hybrid approach activates more input pixels for reconstruction. CPAT [14] proposes a new Channel-Partitioned Attention Transformer to better capture long-range dependencies by sequentially expanding windows along the height and width of feature maps. DRCT [15] centers on stabilizing the information flow and enhancing the receptive fields by incorporating dense-connections within residual blocks, combining the shift-window attention mechanism to adaptively capture global information.

High-performance image super-resolution reconstruction algorithms [10,11,12,13,14,15] have achieved excellent image reconstruction results. However, these methods often involve complex model architectures and a large number of parameters, leading to increased computational costs. They also require substantial memory to store model parameters and intermediate features, which limits their application on memory-constrained devices. Therefore, developing lightweight super-resolution models to balance image reconstruction performance and computational resource consumption is of great significance.

In recent years, researchers have conducted extensive research in the lightweight direction of image super-resolution reconstruction models and achieved excellent results. IMDN [16] enhances feature extraction capabilities by employing an information distillation mechanism to gradually extract hierarchical features, which is effective for super-resolution reconstruction in complex scenes. MADNet [17] captures features at different scales using a multi-scale attention mechanism, thereby improving the model’s ability to perceive details. SwinIR-light [12] is a lightweight version of SwinIR that retains the advantages of the Swin Transformer while reducing model size and computational load, making it suitable for mobile devices. FIWHN [18] proposes a Feature Interaction Weighted Hybrid Network to achieve the model lightweight while reducing the impact of intermediate feature loss on the reconstruction quality.

In this study, we exploit intrinsic features to generate redundant features via low-cost operations [19] and adopt structural re-parameterization [20] to design a lightweight reparameterization layer (LR-Layer). Building upon local feature fusion and local residual learning [21], we propose two efficient depth feature extraction modules called the FS-Block (Feature reuse and structural reparameterization block), LFS Block (Lightweight Feature reuse and Structural reparameterization Block). We assemble efficient image super-resolution networks FS^2^R (Image super-resolution reconstruction network based on feature reuse and structural reparameterization) and FS^2^R-L. Compared with performance-oriented SR methods (e.g., RDN), FS^2^R reduces parameters by 58% and cuts device inference time by 35.8%. Against a typical lightweight baseline (e.g., SwinIR-light), it improves the SSIM [22] on benchmark datasets by 3.24% on average.

The primary contributions of this paper are as follows:We integrate feature reuse and structural reparameterization techniques into image super-resolution reconstruction, resulting in the LR-Layer (Lightweight Reparameterization Layer). By effectively fusing features, this approach significantly reduces the model’s parameter count.We propose efficient deep feature extraction modules, the FS Block (Feature reuse and Structural reparameterization Block) and LFS Block (Lightweight Feature reuse and Structural reparameterization Block), for fast and accurate image super-resolution reconstruction. Under the premise of substantially reducing the model’s parameter count, our method achieves competitive results.We apply the model to IC microscopic image acquisition and conduct inference experiments on an edge platform, obtaining satisfactory results and advancing the practical deployment of image super-resolution.

## 2. Methods

### 2.1. Feature Reuse and Structural Reparameterization

Feature reuse is one of the essential strategies for improving model efficiency and performance in deep learning. By reusing already computed feature maps, redundant calculations can be reduced, thereby alleviating the computational burden of the model. Since multiple layers or operations can share the same feature representation, feature reuse can decrease the number of parameters in the model, enabling it to learn richer and more robust feature representations. Additionally, feature reuse can accelerate the model’s training speed because the model can process data more rapidly. Common methods for feature reuse include residual connections [11,23], dense connections [11,24], feature pyramids [25], and multi-scale feature fusion [10].

To further enhance the performance of neural network models, the scale of networks has been continuously expanding, leading to an increase in the number of features. Consequently, the issue of feature redundancy has gradually garnered attention from developers [19]. This problem is also prevalent in the task of image super-resolution reconstruction, where a large number of similar features exist [26]. Repeatedly obtaining similar features through conventional convolution operations results in significant waste of computational resources. To address this, some researchers have proposed generating a portion of intrinsic features using regular convolutions and then obtaining redundant features via low-cost/cheap operations, such as deep wise convolution and shifting operations (as shown in Figure 1). By concatenating these features, the model can effectively reduce the number of parameters and computational load while ensuring a complete set of output features.

Structural Reparameterization is an optimization technique in deep learning that enhances model performance, efficiency, and generalization by altering the architecture of neural networks [27]. This technique reduces the number of parameters in the model, thereby decreasing its complexity and computational cost. According to Chen et al. [21], Structural Reparameterization typically employs multiple linear operators to generate diverse feature maps during training. These operators are then fused into a single operator through parameter fusion [28], enabling faster inference. The schematic illustration of Structural Reparameterization is shown in Figure 2.

This approach can be employed in deep learning models to increase the number of channels or feature dimensions, thereby enabling subsequent layers to more effectively capture the relationships between different features. The add operation, on the other hand, involves element-wise summation of two tensors to produce a new tensor. This facilitates gradient flow, enhances training stability, and mitigates the vanishing gradient problem in deep networks.

Although neither of the two feature fusion methods introduces additional parameters or FLOPs into the network, experimental validation [21] has shown that under the same batch size, the add operation incurs lower computational costs and shorter runtime compared to the concat operation, as illustrated in Figure 3.

### 2.2. The Reparameterization Layer and Deep Feature Extraction Module

Compared with traditional convolution-activation operations, researchers [19] proposed a method in classification tasks that first uses regular convolutions to generate partial intrinsic features, followed by obtaining redundant features through cost-effective operations. These two types of features are then concatenated using the concat operation. By doing so, the method effectively reduces the model’s parameter count and computational load while ensuring a complete set of output features, as illustrated in Figure 4a. Note that the 1 × 1 convolutions used for channel transformations in Figure 4 are omitted for simplicity.

As described in Section 2.1, compared with the concat operation, the addition operation incurs lower computational costs and has shorter runtime. Chen et al. [21] replaced the concat operation in the classification network module shown in Figure 4a with the addition operation. To comply with the rules of structural reparameterization, the ReLU, which is a non-linear operation, was moved to after the addition operation. Additionally, a batch normalization (BN) operation was introduced in the identity mapping branch to bring non-linearity during training, making the structure more flexible, as illustrated in Figure 4b.

The structure of the LR-Layer used in our image super-resolution reconstruction network is shown in Figure 4c. Compared with Figure 4b, the Batch Normalization operation in the identity mapping branch is replaced with a 1 × 1 convolution. Although batch normalization can reduce the difficulty of network training and prevent overfitting, for the task of image super-resolution reconstruction, the color distribution of the image is normalized after BN, which destroys the original contrast information of the image and thus affects the quality of the super-resolution reconstruction [10]. The 1 × 1 convolution, with its trainable parameters, can learn linear transformations of the input features and improve gradient flow in deep networks [29]. Therefore, it is introduced into the identity mapping branch to address these limitations.

The structure shown in Figure 4c can be reparameterized into the structure depicted in Figure 4d during the inference process, thereby enabling fast inference for the image super-resolution reconstruction network. Our LR-Layer consists only of deep wise convolution (DConv) and activation functions during inference. As a result, it demonstrates superior adaptability to resource limited devices [30].

We replace the traditional convolution-activation operation with the LR-Layer, which is constructed based on feature reuse and structural reparameterization. The complete structure of the LR-Layer is shown in Figure 5.

Building upon dense connectivity, local feature fusion, and local residual learning [11], we design two deep feature extraction blocks, the FS-Block (Feature Reuse and Structural Reparameterization Block) and LFS-Block (Lightweight Feature Reuse and Structural Reparameterization Block), which use the LR-Layer as its basic unit. The structures of the module are illustrated in Figure 6 and Figure 7.

The Contiguous Memory (CM) mechanism is implemented by passing the output features of the preceding FS-Block to each LR-Layer within the current FS-Block. Let $F_{n}$ denote the output of the *n*-th FS-Block, which also serves as the input to the (n+1)-th FS-Block. $F_{n+1}$ denote the output of the (n+1)-th FS-Block. They both consist of G features.

The output of the l-th LR-Layer in the (n+1)-th FS-Block can be expressed as:(1)$$F_{n+1,l}=Act\{W_{n+1,l}\left[Concat\left(F_{n},F_{n+1,1}\ldots ,F_{n+1,l-1}\right)\right]\}$$

In Equation (1), $W_{n+1,l}$ represents the weights of the l-th LR-Layer in the (n+1)-th FS-Block. Suppose $F_{n+1,l}$ consists of $G_{n+1}$ feature maps, its growth rate is $G_{n+1}$ [31]. $F_{n}$, the output features of the n-th FS-Block is concatenated with the output features of the 1st to (l−1)-th LR-Layers of the (n+1)-th FS-Block through the concat operation, resulting in $G_{n}$ + (l−1) × $G_{n+1}$ feature maps. The outputs of the preceding FS-Block and each LR-Layer within the current module are directly connected to all subsequent layers. This structure not only preserves the feedforward nature of the network but also extracts local dense features from the image.

After passing through l LR-Layers in the (n+1)-th FS-Block, the image features require local feature fusion. The output features of the n-th FS-Block module $F_{n}$ and the outputs of the l LR-Layers within the (n+1)-th FS-Block are concatenated via the concat operation and then processed by a 1 × 1 convolution to control the output feature information:(2)$$F_{n+1,LFF}={Conv}_{1\times 1}\left[Concat\left(F_{n},F_{n+1,1},\ldots ,F_{n+1,L}\right)\right]\;\;$$

The number of features in $F_{n+1,\;LFF}$ equals to $F_{n}$. To further enhance the information flow within the network, the FS-Block module introduces local residual learning after local feature fusion. The output of the (n+1)-th FS-Block module $F_{n+1}$ can be represented as:(3)$$F_{n+1}=F_{n}+F_{n+1,LFF}$$

Let $F_{m}$ denote the output of the m-th LFS-Block, which also serves as the input to the (m+1)-th LFS-Block. $F_{m+1}$ denote the output of the (m+1)-th LFS-Block. They both consist of G features.

The output of the l-th LR-Layer in the (m+1)-th LFS-Block can be expressed as:(4)$$F_{m+1,l}=Act\left[W_{m+1,l}(F_{m+1,l-1})\right]$$

In the LFS block, we design a branch bypass to replace the dense connections in the FS block. The dense connection concatenates feature maps from all preceding layers, leading to repeated reuse of early-stage features and high information redundancy. As the network deepens, the channel count increases linearly, causing the parameters of 1 × 1 convolutions to grow quadratically. In contrast, the branch bypass reuses features from adjacent layers for feature propagation, reducing redundancy. With the branch bypass, the network width becomes independent of depth, ensuring that the parameter count grows linearly rather than quadratically.

We define the molecular bypass’s output feature as $F_{x}$. For a LFS Block with L LR-layers, the bypass undergoes L−1 fusion operations.(5)$$F_{x,l-1}={Conv}_{1\times 1}\left[Concat(F_{m+1,l-2},F_{m+1,l-1})\right]$$

After passing through l LR-Layers in the (m+1)-th LFS-Block, the image features require local feature fusion. The output features of the branch bypass $F_{x,L-1}$ and the outputs of the L-th LR-Layer are concatenated via the concat operation and then processed by a 1 × 1 convolution to control the output feature information:(6)$$F_{m+1,LFF}={Conv}_{1\times 1}\left[Concat\left(F_{x,L-1\;},F_{m+1,L-1}\right)\right]$$

The number of features in $F_{m+1,\;LFF}$ equals to $F_{m}$. The LFS-Block module introduces local residual learning after local feature fusion. The output of the (m+1)-th LFS-Block module $F_{m+1}$ can be represented as:(7)$$F_{m+1}=F_{m}+F_{m+1,LFF}$$

### 2.3. The Architecture of the Image Super-Resolution Reconstruction Network Based on Feature Reuse and Structural Reparameterization

The overall architecture of the FS^2^R network is illustrated in Figure 8. The network primarily consists of four components: a shallow feature extraction module, deep feature extraction modules (FS-Blocks), dense feature fusion, and an upsampling module. The input image and output image of the network are represented as $I_{LR}$ and $I_{SR}$. To begin with, shallow feature extraction is performed on the input image $I_{LR}$ using two layers of 3 × 3 convolutions. Specifically, the first 3 × 3 convolution layer is responsible for extracting features $F_{SG}$ from $I_{LR}$.(8)$$F_{SG}={Conv}_{3\times 3}\left(I_{LR}\right)$$

The extracted features $F_{SG}$ are utilized for further shallow feature extraction and global residual learning. The second 3 × 3 convolutional layer takes these features $F_{SG}$ as input to further extract shallow features, and its output features $F_{0}$ serve as the input to the first FS-Block.(9)$$F_{0}={Conv}_{3\times 3}\left(F_{SG}\right)$$

Suppose the FS^2^R network utilizes N FS-Blocks in total, we define the output features of the n-th FS-Block as $F_{n}$, and $F_{n}$ can be formulated as:(10)$$F_{n}=O_{FS\_B,n}(F_{n-1})$$

In Equation (10), $O_{FS\_B,n}$ denotes the operation of the n-th FS-Block. $O_{FS\_B,n}$ is a composite function consisting of deepwise convolution, pointwise convolution, and a non-linear activation function. $F_{n}$ are obtained through internal convolution operations within the n-th FS-Block, and thus $F_{n}$ is referred to as local features.

Dense feature fusion refers to the operation that merges local features from the N FS-Blocks with global residual features $F_{SG}$. In Equation (11), the output features after dense feature fusion are denoted as $F_{DFF}$, $F_{Conv}$ are derived from a sequence of 1 × 1 and 3 × 3 convolutional operations.(11)$$F_{DFF}=F_{Conv}\left[Concat\left(F_{1},\ldots ,F_{n}\right)\right]+F_{SG}$$

Subsequent to acquiring the fused features in the low-resolution space, an upsampling operation is conducted. This can be formally described as:(12)$$I_{SR}=Upsampling(F_{DFF})$$

In image super-resolution reconstruction networks, upsampling operations are one of the core components, used to upscale low-resolution images to the target high-resolution size. Current mainstream upsampling methods can be divided into two major categories: traditional interpolation methods and learnable upsampling methods, each with its own advantages and disadvantages in terms of reconstruction quality, computational efficiency, and high-frequency information retention.

Our understanding of “upsampling from the low-resolution space” is that the upsampling operation itself does not contain any learnable parameters; it is merely a fixed, mathematically rule-based interpolation process entirely executed on the low-resolution feature maps. Traditional interpolation methods, including nearest-neighbor interpolation, bilinear interpolation, and bicubic interpolation, belong to the “upsampling from the low-resolution space” operation. Learnable upsampling operations (deconvolution and sub-pixel convolution) start from the input low-resolution features and are essentially learnable.

Traditional interpolation methods are inherently incapable of introducing new high-frequency information and only perform smooth estimation based on existing pixels, thus causing varying degrees of high-frequency information loss, especially in critical areas such as high-frequency textures and edges.

Commonly used learnable upsampling methods include deconvolution and sub-pixel convolution. Deconvolution is versatile but is prone to the “checkerboard artifacts,” where periodic artifacts appear in the output image. When training is insufficient or the network design is unreasonable, these artifacts can mask the true details of the image, leading to high-frequency information loss or incorrect estimation. Sub-pixel convolution is computationally efficient and is currently widely used in super-resolution networks. The upsampling method used in this paper is sub-pixel convolution, as it performs upsampling at the feature level and can better retain and learn high-frequency details.

## 3. Experimental Verification

### 3.1. Experimental Setup

This study utilizes 800 high-quality RGB images from the DIV2K [32] dataset and 2000 high-quality RGB images from the Flickr2K [33] dataset as the training set. We test the performance of our model on five benchmark datasets: set5 [34], set14 [35], BSD100 [36], Urban100 [37], Manga109 [38]. LR images are generated via bicubic interpolation downsampling. The super-resolution results are assessed using the peak signal-to-noise ratio (PSNR) and structural similarity index (SSIM) [22] calculated on the Y channel of the images in the YCbCr color space.

The size of the LR images is set to 64 × 64. For networks with different scaling factors, the corresponding HR image patches are automatically cropped from the training images. During each training iteration, one HR image patch is cropped from each training image and subjected to data augmentation by randomly applying one of the following operations: 90° rotation, horizontal flipping, or vertical flipping. The image super-resolution network was implemented using the PyTorch2.1.2 deep learning framework and optimized using the Adam optimizer. The initial learning rate for all layers was set to 10^−4^. After 750 training epochs, the learning rate was updated to 10^−5^; after 900 epochs, it was updated to 10^−6^; and the training was terminated after 1000 epochs.

### 3.2. Network Performance Comparison

#### 3.2.1. Quantitative Results

The proposed FS^2^R network in this paper is compared with ten typical image super-resolution reconstruction networks, including VDSR [39], DRCN [40], EDSR-baseline [10], CARN [41], IMDN [16], MADNet [17], SwinIR-light [12], RDN [11], CPAT [14], DRCT [15] and FIWHN [18] using objective evaluation metrics. Table 1, Table 2 and Table 3 show the quantitative comparisons of the super-resolution reconstruction results for scaling factors of ×2, ×3 and ×4, respectively. The best, second-best and third-best results in each table are indicated in bold, underlined and double underlined. It can be observed that our FS^2^R network performs favorably on most datasets, particularly surpassing the majority of models in terms of the Structural Similarity Index Measure (SSIM).

The Structural Similarity Index Measure (SSIM) evaluates the similarity between images based on three relatively independent metrics: luminance, contrast, and structure. The improved performance of our model in this regard implies that it can better capture and restore essential structural details, such as edges and textures. This results in a more natural and visually pleasing outcome that aligns better with human visual perception.

As shown in Table 1, FS^2^R achieves better objective evaluation metrics in the 5 benchmark test sets for the ×2 reconstruction task. Compared with performance-oriented models (such as RDN, CPAT, DRCT), the number of model parameters of FS^2^R is reduced by 58.45%, 55.17%, and 35.31%, respectively. However, the SSIM of FS^2^R (taking BSD100 as an example) is 0.9067, which is higher than that of RDN (0.9017), CPAT (0.9056), and DRCT (0.9051). Compared with other lightweight models (such as CARN, IMDN, FIWHN), the SSIM of FS^2^R-L (taking BSD100 as an example) is increased by 0.97%, 0.77%, and 0.64%, respectively.

As shown in Table 2, FS^2^R achieves better objective evaluation metrics in the 5 benchmark test sets for the ×3 reconstruction task. Compared with performance-oriented models (such as RDN, CPAT, DRCT), the SSIM of FS^2^R (taking BSD100 as an example) is 0.8342, which is higher than that of RDN (0.8093), CPAT (0.8174), and DRCT (0.8182). Compared with other lightweight models (such as CARN, IMDN, FIWHN), the SSIM of FS^2^R-L (taking BSD100 as an example) is increased by 3.81%, 3.65%, and 3.26%, respectively.

As shown in Table 3, compared with performance-oriented models (such as RDN, CPAT, DRCT), the SSIM of FS^2^R (taking BSD100 as an example) is 0.7533, which is higher than that of RDN (0.7419), CPAT (0.7527), and DRCT (0.7532). Compared with other lightweight models (such as CARN, IMDN, FIWHN), the SSIM of FS^2^R-L (taking BSD100 as an example) is increased by 2.48%, 2.42%, and 1.77%, respectively.

The perceptual metrics of the reconstructed images from FS^2^R were compared against those from typical models, with the results presented in Table 4. LPIPS (Learned Perceptual Image Patch Similarity) [42] aligns more closely with human perception than traditional metrics (like PSNR, SSIM). The lower the LPIPS value, the more similar the two images are; conversely, a higher value signifies greater differences.

FS^2^R performs well on most datasets, especially achieving the best results on Set14 and Urban100 (0.1069 and 0.0121), confirming that FS^2^R’s reconstructed image effects are more suitable for human perception.

Different test sets have different data distributions. The BSD100 [36] test set mainly contains images of natural landscapes, animals, plants, and architecture, with relatively rich textures but clear structures. FS^2^R achieved better objective evaluation metrics on BSD100, indicating that the model structure of FS^2^R and the network weights obtained through training are more compatible with the data distribution of BSD100, resulting in superior performance on this dataset. This is manifested in the fact that FS^2^R achieved the highest SSIM scores in tests of various magnification factors. For specific objective metric comparisons, please refer to Table 1, Table 2 and Table 3.

Although the Set5 [34] and Set14 [35] test sets have fewer images, but each contains large numbers of repetitive patterns, sharp edges, and smooth areas. The Urban100 [37] test set contains a large number of urban architectural images with regular and dense geometric structures and long-range continuous edges. Manga109 [38] mainly consists of anime images with large areas of solid colors, clear lines, and minimal natural noise. Compared with FS^2^R, performance-oriented models (such as DRCT [15] and CPAT [14]) can better model long-range dependencies, better maintain the coherence of lines and the purity of solid colors, and perform relatively more stably across various test data distributions. For example, the SSIM metric of FS^2^R is 0.9183 (for the ×4 magnification factor on Manga109 [38]), while under the same conditions, the SSIM metric of DRCT [15] is 0.9304, 1.3% higher than that of FS^2^R; the SSIM metric of CPAT [14] is 0.9309, 1.4% higher than that of FS^2^R. This is one of the reasons why FS^2^R’s objective evaluation metric SSIM is slightly lower than that of performance-oriented models on other test sets.

Performance-oriented models, such as DRCT [15] and CPAT [14], have more parameters and more complex nonlinear transformations, enabling them to learn richer and more detailed image feature representations. These models also include attention mechanisms, which allow the model to adaptively focus on more important areas and allocate more computational resources to reconstructing these important regions. In contrast, FS^2^R and FS^2^R-L, which are designed with engineering applications in mind, strive to find a balance between model size and inference efficiency without adding complex attention structures. Our models aim to achieve good texture recovery and high perceptual evaluation (as shown in Table 4) through relatively simple network architectures. However, when dealing with complex structures that require extremely high precision and coordination, their capabilities are slightly inferior to those of performance-oriented models, and the reconstruction results may contain minor misalignments or blurriness. These minor misalignments or blurriness are detected by the structural similarity index, directly resulting in slightly lower SSIM metrics on test sets other than BSD100 [36] compared to performance-oriented models.

#### 3.2.2. Subjective Evaluation

We carried out image reconstruction experiments at different scaling factors on benchmark datasets. Figure 9 and Figure 10 present the ×4 visual comparisons on the common test datasets. For img_36 from BSD100 [36] and img_88 from Urban100 [37], FS^2^R demonstrates superior grid structure recovery compared to other methods, confirming its effectiveness. As depicted in Figure 9 and Figure 10, FS^2^R’s local reconstruction of BSD100_img_36 and Urban_img_88 achieves results that are on par with or even surpass those of high-performance methods (e.g., RDN) and lightweight methods (e.g., FSRCNN, VDSR, EDSR-baseline, CARN, IMDN and FIWHN). FS^2^R effectively restores edges and textures, making details clearly visible. Notably, in the reconstruction of img_88, FS^2^R accurately recovers the textures of architectural structures. This visual comparison further underscores that FS^2^R reaches an advanced level of performance, meeting the needs of practical engineering applications.

As shown in Table 1, Table 2 and Table 3, FS^2^R attains higher SSIM compared to other models, yet it exhibits relatively lower PSNR metrics. By analyzing the definitions of SSIM and PSNR and conducting an in-depth investigation of the reconstructed images, we have concluded that the image reconstruction performance of FS2R is more focused on enhancing the structural information of images rather than achieving absolute pixel value matching. Research has already demonstrated [23], that in the task of image super-resolution reconstruction, a higher PSNR does not necessarily equate to better image reconstruction quality. Some reconstructed images may have high PSNR values, but their overly smooth details can result in a worse intuitive perception. We conducted edge extraction on the reconstructed images, and the results are displayed in Figure 11. The results show that the reconstructed images of FS^2^R possess richer edge information compared to other models. At the pixel level, these restored details may not be entirely consistent with the original image, which can lead to an increase in the mean squared error (MSE) and a decrease in the PSNR.

Compared to RepGhost [21], which is mainly used for classification tasks, it uses batch normalization layers in the network to enhance feature expression. Through reparameterization techniques, multiple branches are merged into a single convolution to boost inference speed. FS^2^R is mainly used for image reconstruction tasks, replacing BN layers with 1 × 1 convolutions to prevent BN from damaging image contrast information in super-resolution tasks. Compared to RepVGG [20], which is mainly used for image classification, although both networks use reparameterization, FS^2^R also incorporates the lightweight idea of low-cost redundant feature generation, exploring the issue of feature redundancy in neural networks. Compared to GhostSR [26], although both are image super-resolution reconstruction networks, GhostSR only uses feature reuse techniques, while FS2R further compresses the model through structural reparameterization. FS^2^R-L introduces a bypass branch structure to replace dense connections, further reducing information redundancy and parameter growth. The proposed FS^2^R model is a comprehensive improvement on existing lightweight network models, retaining the advantages of advanced models while further exploring the balance between model size and inference efficiency in the engineering applications of neural networks.

#### 3.2.3. Ablation Study

A series of ablation experiments were designed to evaluate the effectiveness of the layers and modules in our model. The network was trained using images of size 64 × 64 and updated with the Adam optimizer, with an initial learning rate of 10^−4^. After 750 training epochs, the learning rate was updated to10^−5^. after 900 epochs, it was updated to 10^−6^, and the training was terminated after 1000 epochs. The ×4 super-resolution reconstruction performance of FS^2^R was evaluated on three benchmark datasets: set14 [35], BSD100 [36] and Urban100 [37]. The experimental results of FS^2^R are documented in Table 5. Experiments revealed that the FS^2^R model achieves optimal performance in ×4 super-resolution reconstruction when employing 16 FS-Blocks, each comprising 8 LR-Layers. Table 6 shows that FS^2^R-L achieves optimal performance in ×4 super-resolution reconstruction when employing 12 LFS-Blocks, each comprising 8 LR-Layers.

In order to further validate the contribution of redundant features generated by various low-cost operations to model performance and to prove the advancement of the structure designed in this paper, we performed substitution tests on low-cost operations in the lightweight layer using the FS^2^R model as the basis. We compared identity mapping, batch normalization, and 1 × 1 convolution (as proposed in this paper). The experimental results indicate that the lightweight layer structure designed in this paper obtains superior results in both objective metrics and subjective evaluation, with the metrics statistics shown in Table 7.

Under the premise of the same model structure, we employ 1 × 1 convolution as a low-cost operation, compared with the use of batch normalization (BN) structure, the average improvement in objective evaluation metrics is 2.72%; compared with identity mapping, the average improvement in objective evaluation metrics is 3.47%. Ablation experiments demonstrate that the model designed in this paper has certain structural innovation and performance advantages in the task of image super-resolution reconstruction.

#### 3.2.4. Inference Time

In engineering applications, in addition to the pursuit of performance of neural network models, the inference time of the model is also an important metric. We selected representative networks and conducted comparative experiments on reconstruction speed using high-performance GPU devices (NVIDIA GeForce RTX 3090) on the BSD100 dataset (×4). The test LR images were of size 64 × 64. An initial model warm-up operation was performed, as the first inference time may include network loading time. Subsequently, each model was repeatedly run 10 times to obtain the average inference time, as shown in Table 8.

While ensuring high-quality image super-resolution reconstruction, our model achieves a better balance between performance and model lightweighting. Compared to high-performance super-resolution networks, on the BSD100 dataset with scale ×4, FS^2^R can obtain similar objective evaluation metrics with reduced parameter counts and even surpass several representative algorithms in specific metrics. Compared to the advanced DRCT, FS^2^R reduces the parameter count by 35% while increasing the SSIM metric by 0.013%. Compared to RDN, FS^2^R reduces the parameter count by 58.5% and increases the SSIM metric by 1.54%. Compared to advanced lightweight networks, our model achieves improved evaluation metrics on certain datasets. Compared to FIWHN, FS^2^R increases the SSIM metric by 1.8%. Compared to SwinIR-light, FS^2^R increases the SSIM metric by 1.71%. Intuitive comparison as shown in Figure 12.

To further validate the performance of FS2R in engineering applications, we conducted edge hardware inference tests using the Jetson Nano B01, with the results shown in Table 9. The Jetson Nano B01 is an embedded and edge computing AI development kit launched by NVIDIA, equipped with a quad-core ARM Cortex-A57 MP core processor and a 128-core NVIDIA Maxwell GPU. Designed specifically for edge computing, it is capable of processing and analyzing data near the data source edge. Its compact size, low power consumption, and powerful computing performance make it highly suitable for deployment in a variety of edge application scenarios.

### 3.3. Evaluation of Datasets in Different Fields

The FS^2^R and FS^2^R-L are image super-resolution reconstruction networks constructed from LR-Layers. In order to validate the generalizability of the network, we performed super-resolution reconstruction network training and image reconstruction experiments using integrated circuit (IC) microscopic images. The training set is composed of REFICS [45] and a portion of self-collected images. REFICS is a large-scale synthetic scanning electron microscope (SEM) dataset, which includes 800,000 SEM images spanning two node technologies: 32 nm and 90 nm. We chose 5000 images with minimal noise and high clarity from the active area, polysilicon, and metal layers in REFICS, and combined them with 2000 self-collected high-definition integrated circuit microscopic images to form the training set. Self-collected high-definition micrographs of IC were acquired from two distinct devices. The first device is fabricated in a 0.18 µm 1P6M BCD (Bipolar-CMOS-DMOS) process; its images were captured at 1800× magnification using an optical electron microscope. The second device is manufactured in a 55 nm 1P5M Bipolar-CMOS technology; its images were obtained at 200,000× magnification via scanning electron microscopy.

We utilized 50 self-collected images that do not overlap with the training set as the overall test set, 30 metal layer images, 30 poly layer images, and 30 DF area images that do not overlap with the training set form the independent test sets. We retrained several typical networks on our IC training dataset for reconstruction performance comparison. The objective performance indicators of our model on ×4 scale are compared with other typical model indicators as shown in Table 10. The visual effects of super-resolution reconstruction of IC microscopic images by FS^2^R-L are demonstrated in Figure 13.

As shown in Table 10, FS^2^R achieves better objective evaluation metrics in the 4 IC image test sets for the ×4 reconstruction task. Compared with performance-oriented models (such as RDN), the structural similarity index of FS^2^R (taking the Overall test set as an example) is 0.9654, which is higher than that of RDN (0.9478). Compared with other lightweight models (such as CARN, IMDN, SwinIR-light), the structural similarity index of FS^2^R-L (taking the Overall test set as an example) is increased by 0.93%, 1.149%, and 1.07%, respectively.

The aim of developing FS^2^R is to significantly lower the hardware requirements and reduce image acquisition costs in the context of IC microscopic image acquisition. The main attention engineers inspect the microscopic structure of integrated circuits is the circuit’s structural characteristics, such as the linewidth and edges within the circuit. The structural similarity measured by SSIM directly meets these requirements. A high SSIM index signifies that the reconstructed IC microscopic image is more in line with the actual situation regarding key structural information, including line shape and edge position. If a model that solely focuses on achieving a high PSNR causes the edges in the microscopic images to be overly smooth, integrating such a model into the IC microscopic image acquisition process would be counterproductive.

In the subjective comparison of reconstructed images between FS^2^R and other models, although FS^2^R’s high SSIM advantage is not very noticeable in natural image test sets (as shown in Figure 9 and Figure 10), in the subjective comparison of IC microscopic image reconstruction, FS^2^R-L effectively restores the edges and textures of the input images, making the details of the integrated circuits clearly visible. In the reconstruction shown in Figure 13a, it is worth highlighting that FS^2^R-L accurately restores the edges where the polysilicon and active regions overlap. This visual comparison further highlights that FS^2^R has achieved an advanced performance level, meeting the requirements of practical engineering applications.

In order to further facilitate the practical deployment and application of FS^2^R in engineering contexts, two approaches were utilized in the task of integrated circuit microscopic image acquisition: standalone acquisition via scanning electron microscopy (SEM) and acquisition via SEM coupled with image super-resolution reconstruction. The acquisition efficiency of these two approaches was statistically compared and shown in Table 11. The super-resolution model employed in the experiment was FS^2^R, the edge device utilized was Jetson Nano B01, and the acquisition output image size was 256 × 256.

When the same electron beam dwell time is employed, for the same target, the time needed for the SEM combined with super-resolution reconstruction acquisition mode is decreased by 79.3% compared to the time needed for acquiring microscopic images using the SEM alone. To further improve the quality of microscopic image acquisition, when our acquisition method uses a dwell time of 100 microseconds per pixel, the acquisition time can be saved by 73.1%.

## 4. Conclusions

In response to the challenges of high computational complexity and large memory consumption in current image super-resolution networks, we propose FS^2^R and FS^2^R-L, networks that leverage feature reuse and structural reparameterization techniques. FS^2^R and FS^2^R-L enable fast and accurate extraction of local and global deep features from images. By leveraging intrinsic features and low-cost operations to generate redundant features, we design a reparameterization layer (LR-Layer) for feature reuse. Additionally, we develop the FS-Block and LFS-Block module. These components collectively form the FS^2^R and FS^2^R-L network. Experimental results indicate that FS^2^R achieves a good balance between performance and network complexity compared with other state-of-the-art algorithms. However, the reconstructed images still exhibit some edge blurring and artifacts. Compared to advanced lightweight image super-resolution reconstruction algorithms, there is still a large compression space for the model size. We will continue to improve the compression of our model. In the future, we will focus on improving the image quality of the reconstructed results while further advancing the lightweight design of the model to better adapt it for resource-limited devices.

## Figures and Tables

**Figure 1 sensors-25-05989-f001:**
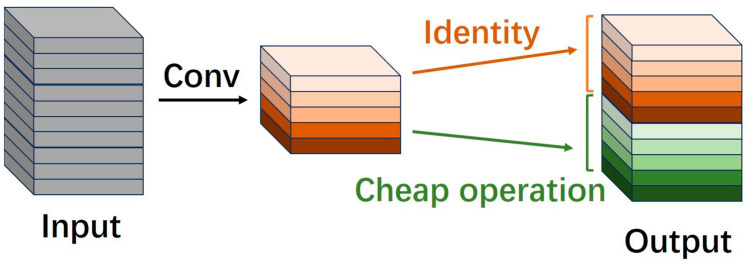
Obtain redundant features with cheap operation.

**Figure 2 sensors-25-05989-f002:**
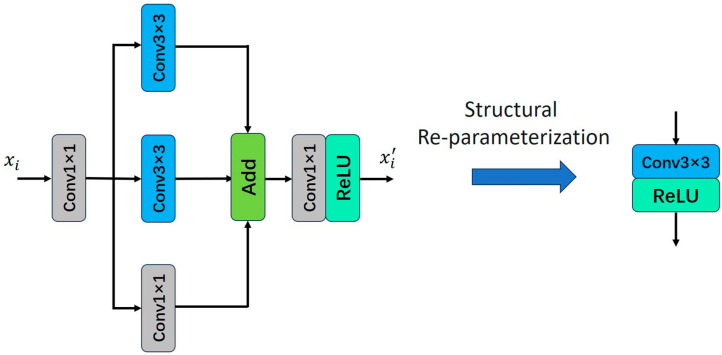
Parallel structure for structural reparameterization.

**Figure 3 sensors-25-05989-f003:**
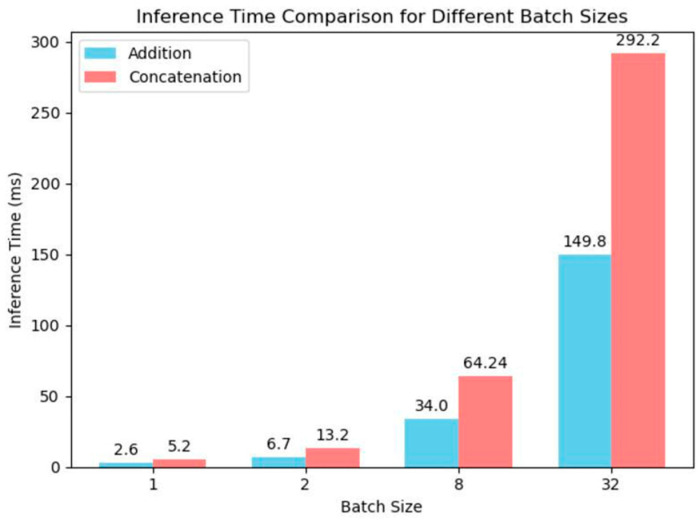
Comparison of inference time between concatenation and addition operations [21].

**Figure 4 sensors-25-05989-f004:**
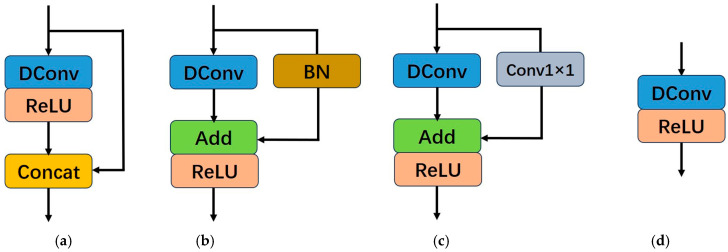
The evolution process of LR-Layer for image super-resolution reconstruction. (**a**) is the basic layer of GhostNet for classification tasks. (**b**) is the basic layer of RepGhost for classification tasks. (**c**) is the layer that we designed for image super resolution task. (**d**) is the structure after reparameterization operation.

**Figure 5 sensors-25-05989-f005:**
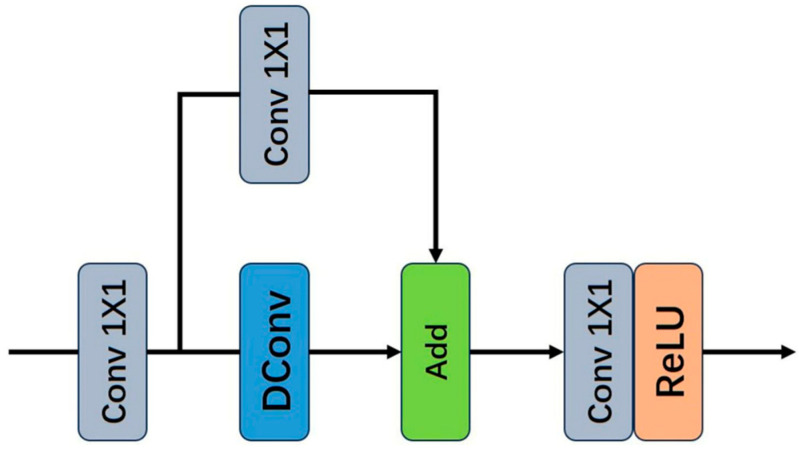
Complete LR-Layer structure.

**Figure 6 sensors-25-05989-f006:**
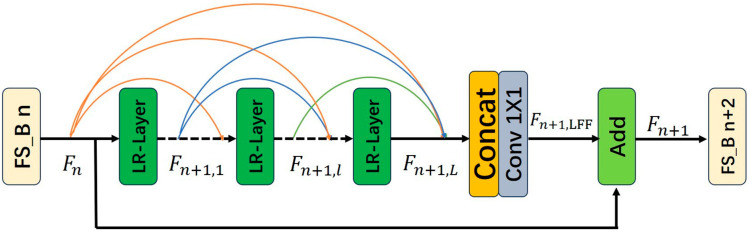
Structure of Feature Reuse and Structural Reparameterization Block.

**Figure 7 sensors-25-05989-f007:**
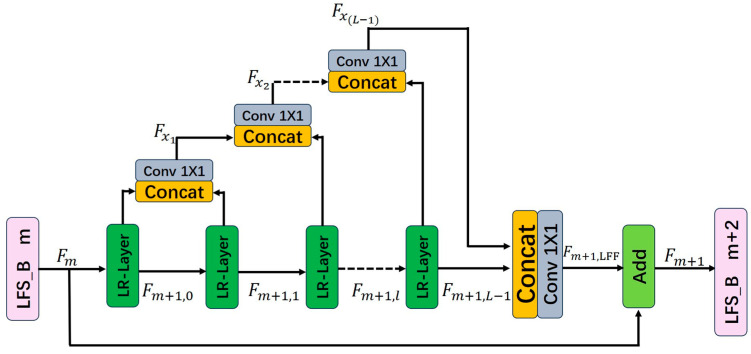
Structure of Lightweight Feature Reuse and Structural Reparameterization Block.

**Figure 8 sensors-25-05989-f008:**
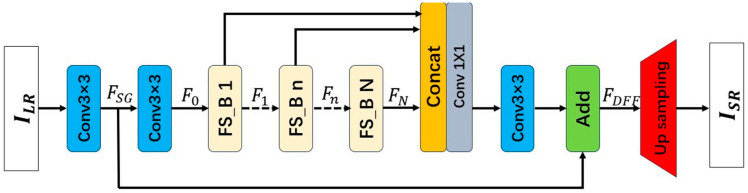
Structure of Image super-resolution reconstruction network based on feature reuse and structural reparameterization.

**Figure 9 sensors-25-05989-f009:**
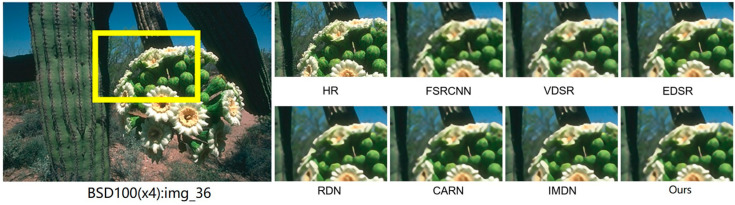
Reconstruction results comparison of typical algorithms and FS^2^R on BSD100 dataset with scale factor of 4.

**Figure 10 sensors-25-05989-f010:**
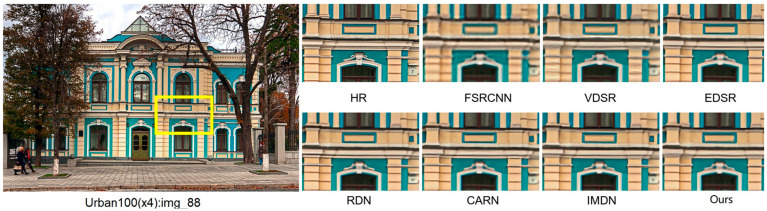
Reconstruction results comparison of typical algorithms and FS^2^R on Urban100 dataset with scale factor of 4.

**Figure 11 sensors-25-05989-f011:**
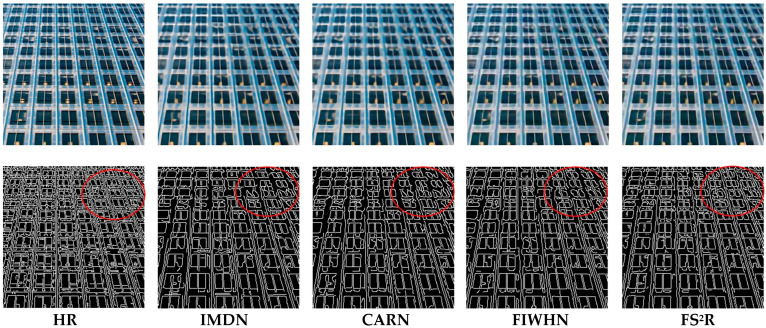
Edge extraction on the reconstructed image of typical algorithms and FS^2^R on Urban100 (img_30) with scale factor of 4.

**Figure 12 sensors-25-05989-f012:**
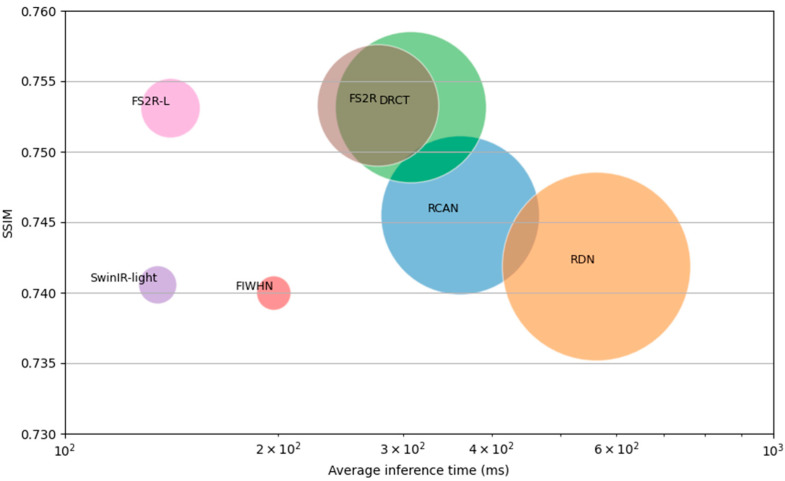
Comparison of our model with other models in terms of performance, parameter count, and inference time.

**Figure 13 sensors-25-05989-f013:**
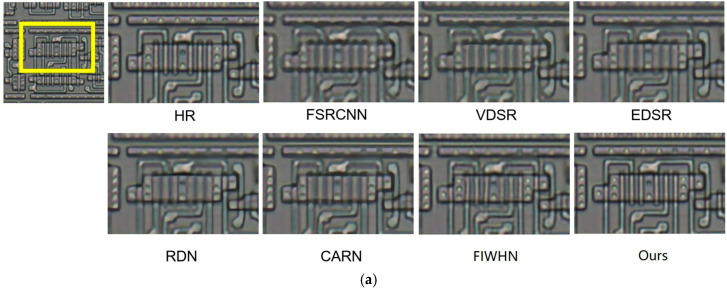
Reconstruction results comparison of typical algorithms and FS^2^R-L on IC dataset with scale factor of 4 ((**a**) shows the reconstruction result of the integrated circuit polysilicon layer obtained through an optical microscope; (**b**) shows the reconstruction result of the integrated circuit active area obtained through a scanning electron microscope.).

**Table 1 sensors-25-05989-t001:** Quantitative comparison of different algorithms on five benchmark datasets with scale factor of 2. The best, second-best and third-best results in each table are indicated in bold, under-lined and double underlined.

Model	Scale	Set5 [34]	Set14 [35]	BSD100 [36]	Urban100 [37]	Manga109 [38]
		PSNR/SSIM	PSNR/SSIM	PSNR/SSIM	PSNR/SSIM	PSNR/SSIM
VDSR [39]	×2	37.53/0.9587	33.03/0.9124	31.90/0.8960	30.76/0.9140	37.22/0.9750
DRCN [40]		37.63/0.9588	33.04/0.9118	31.85/0.8942	30.75/0.9133	37.55/0.9732
EDSR-baseline [10]		37.99/0.9604	33.57/0.9175	32.16/0.8994	31.98/0.9272	38.54/0.9769
CARN [41]		37.76/0.9590	33.52/0.9166	32.09/0.8978	31.92/0.9256	38.36/0.9765
IMDN [16]		38.00/0.9605	33.63/0.9177	32.19/0.8996	32.17/0.9283	38.88/0.9774
MADNet [17]		37.85/0.9600	33.38/0.9161	32.04/0.8979	31.62/0.9233	-
SwinIR-light [12]		38.14/0.9611	33.86/0.9206	32.31/0.9012	32.76/0.9340	39.12/0.9783
RDN [11]		38.24/0.9614	34.01/0.9212	32.34/0.9017	32.89/0.9353	39.18/0.9780
CPAT [14]		**38.68**/0.9633	**34.91**/**0.9277**	**32.64**/0.9056	**34.76**/**0.9481**	40.28/**0.9814**
DRCT [15]		38.62/0.9628	34.84/0.9272	32.62/0.9051	34.44/0.9464	**40.31**/0.9804
FIWHN [18]		38.16/0.9613	33.73/0.9194	32.27/0.9007	32.75/0.9337	39.07/0.9782
FS^2^R (Ours)		37.83/**0.9637**	33.33/0.9227	32.04/**0.9067**	31.27/0.9245	38.79/0.9789
FS^2^R-L (Ours)		37.79/0.9631	33.30/0.9221	31.97/0.9065	31.22/0.9240	38.71/0.9781

**Table 2 sensors-25-05989-t002:** Quantitative comparison of different algorithms on five benchmark datasets with scale factor of 3. The best, second-best and third-best results in each table are indicated in bold, under-lined and double underlined.

Model	Scale	Set5 [34]	Set14 [35]	BSD100 [36]	Urban100 [37]	Manga109 [38]
		PSNR/SSIM	PSNR/SSIM	PSNR/SSIM	PSNR/SSIM	PSNR/SSIM
VDSR [39]	×3	33.66/0.9213	29.77/0.8314	28.82/0.7976	27.14/0.8279	32.01/0.9340
DRCN [40]		33.82/0.9226	29.76/0.8311	28.80/0.7963	27.15/0.8276	32.24/0.9343
EDSR-baseline [10]		34.37/0.9270	30.28/0.8417	29.09/0.8052	28.15/0.8527	33.45/0.9439
CARN [41]		34.29/0.9255	30.29/0.8407	29.06/0.8034	28.06/0.8493	33.50/0.9440
IMDN [16]		34.36/0.9270	30.32/0.8417	29.09/0.8046	28.17/0.8519	33.61/0.9445
MADNet [17]		34.16/0.9253	30.21/0.8398	28.98/0.8023	27.77/0.8439	-
SwinIR-light [12]		34.62/0.9289	30.54/0.8463	29.20/0.8082	28.66/0.8624	33.98/0.9478
RDN [11]		34.71/0.9296	30.57/0.8468	29.26/0.8093	28.80/0.8653	34.13/0.9484
CPAT [14]		35.16/0.9334	31.15/0.8557	29.56/0.8174	**30.52**/**0.8923**	35.66/0.9559
DRCT [15]		**35.18**/**0.9338**	**31.24**/**0.8569**	**29.68**/0.8182	30.34/0.8910	**35.76**/**0.9575**
FIWHN [18]		34.50/0.9283	30.50/0.8451	29.19/0.8077	28.62/0.8607	33.97/0.9472
FS^2^R (Ours)		34.17/0.9329	30.16/0.8531	29.55/**0.8342**	28.78/0.8733	33.56/0.9484
FS^2^R-L (Ours)		34.16/0.9331	30.11/0.8519	29.55/0.8340	28.73/0.8731	33.51/0.9481

**Table 3 sensors-25-05989-t003:** Quantitative comparison of different algorithms on five benchmark datasets with scale factor of 4. The best, second-best and third-best results in each table are indicated in bold, under-lined and double underlined.

Model	Scale	Set5 [34]	Set14 [35]	BSD100 [36]	Urban100 [37]	Manga109 [38]
		PSNR/SSIM	PSNR/SSIM	PSNR/SSIM	PSNR/SSIM	PSNR/SSIM
VDSR [39]	×4	31.35/0.8838	28.01/0.7674	27.29/0.7251	25.18/0.7524	28.83/0.8870
DRCN [40]		31.53/0.8854	28.02/0.7670	27.23/0.7233	25.14/0.7510	28.93/0.8854
EDSR-baseline [10]		32.09/0.8938	28.58/0.7813	27.57/0.7357	26.04/0.7849	30.35/0.9067
CARN [41]		32.13/0.8937	28.60/0.7806	27.58/0.7349	26.07/0.7837	30.47/0.9084
IMDN [16]		32.21/0.8948	28.58/0.7811	27.56/0.7353	26.04/0.7838	30.45/0.9075
MADNet [17]		31.95/0.8917	28.44/0.7780	27.47/0.7327	25.76/0.7746	-
SwinIR-light [12]		32.44/0.8976	28.77/0.7858	27.69/0.7406	26.47/0.7980	30.92/0.9151
RDN [11]		32.47/0.8990	28.81/0.7871	27.72/0.7419	26.61/0.8028	31.00/0.9151
CPAT [14]		**33.19**/**0.9069**	29.34/**0.7991**	28.04/0.7527	28.22/**0.8408**	**32.69**/**0.9309**
DRCT [15]		33.11/0.9064	**29.35**/0.7984	**28.18**/0.7532	**28.06**/0.8378	32.59/0.9304
FIWHN [18]		32.30/0.8967	28.76/0.7849	27.68/0.7400	26.57/0.7989	30.93/0.9131
FS^2^R (Ours)		32.05/0.9023	28.57/0.7970	27.57/**0.7533**	25.89/0.7918	30.92/0.9183
FS^2^R-L (Ours)		32.01/0.9020	28.56/0.7971	27.55/0.7531	25.87/0.7921	30.90/0.9181

**Table 4 sensors-25-05989-t004:** Perceptual metrics (LPIPS) quantitative comparison of different algorithms on benchmark datasets with scale factor of 4. The best, second-best and third-best results in each table are indicated in bold, under-lined and double underlined.

Model	Scale	Set5 [34]	Set14 [35]	BSD100 [36]	Urban100 [37]
DRCT [15]	×4	0.2748	0.2434	0.3373	0.1576
IMDN [16]		**0.1322**	0.1179	0.1964	0.0144
FIWHN [18]		0.1871	0.1137	**0.1907**	**0.0121**
RepRFN [43]		0.2487	0.1221	0.1937	0.0215
FS^2^R (Ours)		0.1603	**0.1069**	0.2056	**0.0121**

**Table 5 sensors-25-05989-t005:** Comparison of model parameter quantity and performance for different numbers of FS-Block and LR-Layer on FS2R. The best results in the table are indicated in bold.

Block	Layer	Parameter	Set14	BSD100	Urban100
			PSNR/SSIM	PSNR/SSIM	PSNR/SSIM
12	6	4.94 M	28.13/0.7936	27.51/0.7506	25.75/0.7866
	8	6.95 M	28.52/0.7959	27.54/0.7522	25.80/0.7888
16	6	6.86 M	28.53/0.7961	27.55/0.7525	25.86/0.7887
	8	9.14 M	**28.57/0.7970**	**27.57/0.7533**	**25.89/0.7918**

**Table 6 sensors-25-05989-t006:** Comparison of model parameter quantity and performance for different numbers of LFS-Block and LR-Layer on FS2R-L. The best results in the table are indicated in bold.

Block	Layer	Parameter	Set14	BSD100	Urban100
			PSNR/SSIM	PSNR/SSIM	PSNR/SSIM
12	6	1.65 M	28.46/0.7951	27.50/0.7524	25.71/0.7885
	8	2.19 M	**28.5** **6/0.797** **1**	**27.5** **5/0.753** **1**	**25.8** **7/0.79** **21**
16	6	2.08 M	28.51/0.7963	27.51/0.7526	25.73/0.7892
	8	3.01 M	28.43/0.7923	27.46/0.7519	25.71/0.7883

**Table 7 sensors-25-05989-t007:** Comparison of objective indicators of reconstruction effect when using different low-cost operations with the same model structure. The best results in the table are indicated in bold.

Identity	BN	Conv 1 × 1	Set5 [34]	Set14 [35]	BSD100 [36]
			PSNR/SSIM	PSNR/SSIM	PSNR/SSIM
√	×	×	32.02/0.9011	28.54/0.7963	26.99/0.7381
×	√	×	31.16/0.8893	27.96/0.7831	26.07/0.6802
×	×	√	**32.05**/**0.9023**	**28.57**/**0.7970**	**27.57**/**0.7533**

**Table 8 sensors-25-05989-t008:** Comparison of model parameter quantity and performance on high-performance GPU devices. The best results in the table are indicated in bold. The best, second-best and third-best results in each table are indicated in bold, under-lined and double underlined.

Model	Parameter	SSIM	Average Inference Time
RCAN [44]	15,592 K	0.7455	360.9 ms
RDN [11]	22,000 K	0.7419	561.4 ms
DRCT [15]	14,130 K	0.7532	280.5 ms
FIWHN [18]	**725 K**	0.7400	197 ms
SwinIR-light [12]	897 K	0.7406	**134.9 ms**
FS^2^R	9140 K	**0.7533**	276.3 ms
FS^2^R-L	2190 K	0.7531	140.7 ms

**Table 9 sensors-25-05989-t009:** Comparison of model parameter quantity and performance on edge hardware. The best results in the table are indicated in bold. The best and second-best results in each table are indicated in bold and under-lined.

Model	Parameter	SSIM	Average Inference Time
RCAN [44]	15,592 K	0.7395	589.1 ms
RDN [11]	22,000 K	0.7401	636.7 ms
IMDN [16]	715 K	0.7312	**265.4 ms**
FS^2^R	9140 K	**0.7** **513**	473.5 ms
FS^2^R-L	2190 K	0.7490	319.2 ms

**Table 10 sensors-25-05989-t010:** Quantitative comparison of different algorithms on IC test sets with scale factor of 4. The best, second-best and third-best results in each table are indicated in bold, under-lined and double underlined.

Model	Scale	Overall	Metal	DF	Poly
		PSNR/SSIM	PSNR/SSIM	PSNR/SSIM	PSNR/SSIM
FSRCNN	×4	36.56/0.9358	37.97/0.9630	34.21/0.8870	32.12/0.8930
CARN		38.31/0.9523	38.87/0.9763	36.77/0.9411	32.45/0.9167
EDSR-baseline		38.19/0.9496	38.01/0.9281	34.24/0.8943	31.43/0.9156
IMDN		37.97/0.9471	41.99/0.9712	38.00/0.9488	32.06/0.9131
SwinIR-light		**38.71**/0.9510	40.37/0.9752	38.21/0.9551	32.31/0.9148
RDN		38.33/0.9478	**42.27**/0.9757	**38.68**/0.9620	**33.05**/0.9243
FS^2^R (Ours)		38.38/**0.9654**	41.19/**0.9780**	37.92/**0.9668**	32.62/**0.9255**
FS^2^R-L (Ours)		38.32/0.9612	41.03/0.9758	37.63/0.9632	32.51/0.9240

**Table 11 sensors-25-05989-t011:** Comparison of the efficiency between standalone acquisition via SEM and acquisition via SEM coupled with image super-resolution reconstruction.

	SEM	SEM + SR	
Final Image Size	256 × 256	256 × 256	
SEM Image Size	256 × 256	64 × 64	
Beam Dwell Time	50 μs/pixel	50 μs/pixel	100 μs/pixel
Acquisition Time	3276.8 ms	204.8 ms	409.6 ms
SR ×4 Runtime	—	473.5 ms	
Total Time	3276.8 ms	678.3 ms	883.1 ms

## Data Availability

The original contributions presented in this study are included in the article. Further inquiries can be directed to the corresponding author(s).

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
