# Peer review of "Image Super-Resolution Reconstruction Network Based on Structural Reparameterization and Feature Reuse"

_sensors, 2025, doi:10.3390/s25195989_

Round 1
Reviewer 1 Report
Comments and Suggestions for Authors
The authors proposed image super-resolution reconstruction networks based on feature reuse and structural reparameterization techniques to improve performance of image super-resolution reconstruction, ensuring that the networks maintain reconstruction performance while being more suitable for deployment in resource-limited environments. However, there are some problems as follows:
1. The authors need to expand the experimental analysis section, consider adding ablation studies to quantify the contributions of the various components in the network.
2. The scheme of Structural Reparameterization in Section 2.1,LR-Layer in Section 2.2 used very simple network structure compared with other methods, so I wonder the innovativeness of this method because this method sacrificed a lot of SR quality in order to improve efficiency.
3. The metric values in Table 1, 2, 3 and the subjective comparison images did not demonstrate advantages compared with other methods. The authors need to provide reasons for its weaker performance in some metrics. For each metric, a detailed discussion of its significance and an explanation of the results should be provided.
4. In Equation (12), which upsampling operation is conducted from the low-resolution space? The authors need to clarify whether the general upsampling operation will lead to the loss of high-frequency details.
Reviewer 2 Report
Comments and Suggestions for Authors
The paper present a super resolution network (FS2R/FS2R-L) aimed at integrated circuit (IC) microscopic images. The method combines feature reuse and structural reparameterization to reduce parameters and speed up inference.
The method is tested on common SR datasets of natural images. The results over these datasets show that FS2R achieves incremental, but competitive, SSIM and comparable PSNR to existing lightweight approaches.
The novelty is limited as RepGhost, RepVGG, and related works already apply structural reparameterization. The contribution seems like an incremental adaptation.
Some Figs (1,2,3,4,7) are very low resolution (visible JPEG artifacts, poor crops). This limits readability and makes it difficult to appreciate results.
The central motivation of the paper is deployment in integrated circuit microscopic imaging, but the experimental validation in this domain is very limited. The authors only present qualitative results (section 3.3) on 50 self-collected test images (Figure 12) and no actual measures. Maybe a supplementary task (detection of defects on both the LR and Upscaled images) could validate the approach, as motivated in the Introduction section.
For a revision I suggest to better clarify the novelty relative to RepGhost/RepVGG/GhostSR on top of swapping BN with a 1×1 conv for SR.
Replot figures as vector graphics (if .jpg is required by the editorial system save them at an higher res), and improve qualitative comparisons with proper zoom-ins and baselines.
Provide some quantitative evaluation on the IC dataset, not just two qualitative images.
Maybe include perceptual metrics (LPIPS,) alongside PSNR/SSIM for the natural images as that would further show how the model compares in this task.
Reviewer 3 Report
Comments and Suggestions for Authors
The article is devoted to the current topic of super-resolution image reconstruction. The authors propose an approach to forming the image reconstruction network based on local feature fusion and local residual learning. The described network has fewer parameters and a higher operating speed. This makes it possible to use it in systems with limited computing resources.
The article describes the proposed approach in sufficient detail. The experimental results confirm its effectiveness. The approach is based on new findings in the field of deep learning. The references contains quite a lot of new publications and corresponds to the topic of the article.
The article as a whole makes a positive impression. However, it does not pay enough attention to the practical aspects related to the use of the proposed network in systems with limited computing resources. At the same time, it is declared that the approach is aimed specifically at such systems. It would be good to describe the characteristics of such devices and what effect is achieved for them in this case. At the same time, it may make sense to consider this task in combination with other tasks and show the overall gain.
This remark is of a recommendatory nature and does not reduce the overall positive assessment of the work. I believe that the article can be recommended for publication.
Round 2
Reviewer 1 Report
Comments and Suggestions for Authors
The authors only take BSD100 as an example for all the comparisons, but I suggest the reasons of lower SSIM values than other methods, such as CPAT [14] for other databases should be explained.
